# MATCHING RECEPTOR TO ODORANT WITH PROTEIN LANGUAGE AND GRAPH NEURAL NETWORKS

**Matej Hladiš**[*]**, Maxence Lalis, Sébastien Fiorucci, Jérémie Topin**[*]
Institut de Chimie de Nice, Université Côte d'Azur, UMR 7272 CNRS, France
`{matej.hladis,jeremie.topin}@univ-cotedazur.fr`

## ABSTRACT

Odor perception in mammals is triggered by interactions between volatile organic compounds and a subset of hundreds of proteins called olfactory receptors (ORs). Molecules activate these receptors in a complex combinatorial coding allowing mammals to discriminate a vast number of chemical stimuli. Recently, ORs have gained attention as new therapeutic targets following the discovery of their involvement in other physiological processes and diseases. To date, predicting molecule-induced activation for ORs is highly challenging since 43% of ORs have no identified active compound. In this work, we combine [CLS] token from protBERT with a molecular graph and propose a tailored GNN architecture incorporating inductive biases from the protein-molecule binding. We abstract the biological process of protein-molecule activation as the injection of a molecule into a protein-specific environment. On a newly gathered dataset of 46 700 OR-molecule pairs, this model outperforms state-of-the-art models on drug-target interaction prediction as well as standard GNN baselines. Moreover, by incorporating non-bonded interactions the model is able to work with mixtures of compounds. Finally, our predictions reveal a similar activation pattern for molecules within a given odor family, which is in agreement with the theory of combinatorial coding in olfaction.

## 1 INTRODUCTION

Mammalian sense of smell constantly provides information about the composition of the volatile chemical environment and is able to discriminate thousands of different molecules. At the atomic scale, volatile organic compounds are recognized by specific interactions with protein receptors expressed at the surface of olfactory neurons (Buck & Axel, 1991). Mammalian epithelium expresses hundreds of different olfactory receptors (ORs), belonging to the G protein-coupled receptors (GPCRs), which constitute the largest known multigene family (Niimura & Nei, 2003). The recognition of odorants by ORs is based on the complementarity of structures and hydrophobic or van der Waals interactions which leads to low molecular affinity (Katada et al., 2005). With the exception of a few conserved amino acids, the sequences of ORs show little identity. In particular, the ligand-binding pocket has hypervariable residues (Pilpel & Lancet, 1999) that are relatively well conserved between orthologs. This property gives ORs the ability to bind a wide variety of molecules that differ in structure, size, or chemical properties. The recognition of odorants is done according to the combinatorial code of activation (Malnic et al., 1999). Each odorant is recognized by several ORs, whereas an individual OR can bind several odorants with distinct affinities and specificities (Zhao et al., 1998). This combinatorial code is sensitive to subtle modifications, so the response of a single receptor can have a major influence on the smell perception. Even a small sequence modification could affect odorant responsiveness (Keller et al., 2007; Mainland et al., 2014). On the other hand, structural and functional modifications of an odorant can abolish the interaction with a specific receptor (Katada et al., 2005), and even lead to a different smell perception (Sell, 2006). So far the combinatorial code of the majority of odorants remains unknown. Identifying the recognition spectrum of each OR is therefore essential to decipher the mechanisms of the olfactory system and subsequently build models capable of cracking the combinatorial code of activation.

---

[*]Corresponding authors.

There is only a limited number of models designed to match ligands and ORs. Namely Kowalewski & Ray (2020) follow a molecule-oriented approach and predict agonists for a subset of 34 ORs (24 wild types and 10 variants) by representing molecules via fingerprints (Morgan, 1965; Klekota & Roth, 2008) and building an individual SVM model for each OR. On the other hand, Cong et al. (2022) focus on receptors and address a more complex problem of predicting active ORs for 4 given molecules. In their random forest model, each amino acid of the protein sequence is described by 3 physico-chemical properties and the molecules by a subset of Dragon descriptors (Mauri et al., 2006). In a more general approach, Gupta et al. (2021) consider any OR-molecule pair and use BiLSTM (Graves & Schmidhuber, 2005) to predict the binding of a molecule, represented by SMILES string, to an OR sequence.

In contrast, in this work we use a graph and a [CLS] token embedding from protBERT (Elnaggar et al., 2021) to represent molecules and receptors, respectively. We abstract receptor-molecule binding as the injection of a molecule into a protein specific environment. This is achieved by using molecular topology as a layout for the message passing process and copying the protein representation to each node of the molecular graph. As a result, the redundancy of protein information enables a local processing of the "protein environment" and achieves better performance than a common strategy of processing the receptor and the molecule in parallel. This abstraction leads to a graph with the number of nodes depending only on the size of the molecule.

Molecules with flexible moieties can undergo conformational changes upon binding to maximize interactions with the receptor binding cavity. This structural adaptation modifies the strength of internal non-bonded forces. However, standard GNN architectures are restricted to the molecular topology. Thus, we have built a tailored GNN architecture combining local interaction of bonded atoms, as done by standard GNNs, with multi-head attention, giving the model the ability to incorporate interactions between any pair of atoms. We show that this architecture outperforms other baselines as well as previous work on olfactory receptor-molecule activation prediction.

Finally, we found a relationship between human odor perception and model predictions, strengthening the biological relevance of the model. The results for humans are in full agreement with the experimental work done by Nara et al. (2011). By analyzing the predictions for human ORs, we observe that the combinatorial codes exhibit large diversity. The OR repertoire contains mostly narrow receptors with several broadly-tuned ones. The results also highlight the existence of odor-specific receptors, but most odors are coded in a complex activation pattern.

## 2 PRELIMINARIES

### 2.1 PROTEIN LANGUAGE MODELS

Recently, protein language models emerged as unsupervised structure learners (Rao et al., 2021b; Vig et al., 2021; Rives et al., 2021), allowing to extract abstract vector representations of proteins. As in natural language processing (NLP), large models with millions of parameters (e.g BERT (Devlin et al., 2019)) are trained on vast databases of amino acid sequences (Steinegger et al., 2019; Steinegger & Söding, 2018; UniProt Consortium, 2019; Suzek et al., 2007). Rao et al. (2019) trained and evaluated various natural language processing models on a set of structurally relevant tasks. Elnaggar et al. (2021) went further and trained a list of powerful NLP models ranging from 200M to 11B parameters. Recently proposed MSA Transformer (Rao et al., 2021a) exploits evolutionary relationships by using Multiple Sequence Alignment (MSA) as input rather than a single protein sequence at a time. AlphaFold2 (Jumper et al., 2021; Evans et al., 2021) extends the idea of using MSA and combines it with an experimentally obtained protein template in an end-to-end model trained on supervised structure prediction.

### 2.2 GRAPH NEURAL NETWORKS

In recent years, graph neural networks (Kipf & Welling, 2016; Gilmer et al., 2017; Simonovsky & Komodakis, 2017; Veličković et al., 2018; Wang et al., 2018; Zhou et al., 2018; Battaglia et al., 2018) have grown rapidly in popularity and received considerable attention in various domains such as drug design (Torng & Altman, 2019), physics (Shlomi et al., 2021), and chemistry (Gilmer et al., 2017; Yang et al., 2019; Gasteiger et al., 2020b;a). Chemistry is a particularly promising field for GNN applications since a molecule can be naturally represented as a graph $G = \{\mathcal{V}, \mathcal{E}\}$, where $\mathcal{V}$ is the set

of nodes (atoms) and $\mathcal{E}$ the set of edges (bonds). Each node $v \in \mathcal{V}$ and each edge $(u,v) \in \mathcal{E}$ in $G$ have an initial vector of features, $x_v$ and $e_{u,v}$, respectively. These feature vectors contain information about atom/edge properties such as atomic number, atomic weight, or bond type.

## 3 MODEL

To tackle the problem of receptor-molecule activation prediction we take a proteo-chemometric approach (Qiu et al., 2017) where the model estimates the probability $P(m,r)$ that a given molecule $m$ activates a given receptor $r$. The input to the model is the amino acid sequence of the receptor on one side and the molecular graph on the other side. See Fig. 1 (a) for the model outline.

Protein sequence representation is obtained using protBERT (Elnaggar et al., 2021) pretrained on a dataset with 217M protein sequences (Suzek et al., 2007). Olfactory receptors are biopolymers of around 300 amino acids and modeling each amino acid explicitly is computationally demanding. To keep the costs relatively low, we take the advantage of the BERT's classification token ([CLS]) (Devlin et al., 2019) which aggregates information about an entire protein sequence. This way we model amino acids implicitly as they contribute to the [CLS] token embedding, and at the same time, the sequence representation is reduced from roughly $300 \times d$ to only $d$ , where $d$ is an embedding dimension. We concatenate the [CLS] token embedding from the last 5 layers of protBERT and use this vector as the receptor input. A molecule is represented as a graph $G$ with node and edge features in Tab. 5. Note that the representation relies solely on the molecular topology and does not take into account spatial characteristics.

The model architecture itself was designed by observing several inductive biases which emerge in molecule-receptor interactions:

**Locality.**   The binding between a receptor and a molecule relies on subtle interactions between the binding groups of both protagonists. In particular, molecules contain functional groups that are crucial for forming intermolecular bonds with the receptor. Since we use an aggregated sequence representation, and thus do not explicitly model interactions between amino acids and atoms in the molecule, we emulate this "locality" by copying the receptor embedding to each node of the molecular graph (Fig. 1 (a)). This way the sequence representation is part of the node embedding and can be locally changed based on the connectivity and neighborhood properties of each node.

**Non-bonded interactions.**   In addition to bonded interactions, atoms in the same molecule interact with each other via electrostatic and hydrophobic interactions. To account for these, we use the attention mechanism (Vaswani et al., 2017), which allows a flow of information between any pair of atoms (Fig. 1 (b)). This is particularly relevant for flexible molecules where two distant atoms in the topology could be close in the conformation. Such approach also inherently takes into account interactions between multiple molecules and thus allows for modeling mixtures.

**Receptor-molecule complex.**   When a molecule binds to a receptor, the two form a stable complex that can be considered as a single entity. To accommodate this observation, we combine the molecule and receptor inputs early in the model pipeline rather than mixing them at a late stage. This facilitates the flow of information between them throughout the model.

Combining the above observations, we propose the model architecture in Fig. 1. Based on the *locality* and *receptor-molecule complex* observations, we inject receptor embedding into the molecular graph as additional node features, and we use the molecular topology as a layout for the message passing process (Fig. 1 (a)). This approach reduces the biological problem of protein-molecule interaction to a small-scale graph-level binary classification where a molecule is injected into a receptor environment. In this formulation, the size of the graph depends solely on the number of atoms in the molecule.

Given the above abstraction, any standard GNN architecture might be applied to predict a molecule-receptor activation. However, standard models do not allow for information to pass outside of the graph topology. To account for *non-bonded interactions*, we combine the local processing done by a GNN with the multi-head attention. To enhance the expressive power of the network, we use two separate identical GNNs for queries and keys/values. We denote these GNNs Q-MPNN for "queries" and KV-MPNN for "keys/values". Each row in the queries and keys/values is a node embedding resulting from the corresponding GNN (Fig. 1 (b)). Then we use the multi-head attention

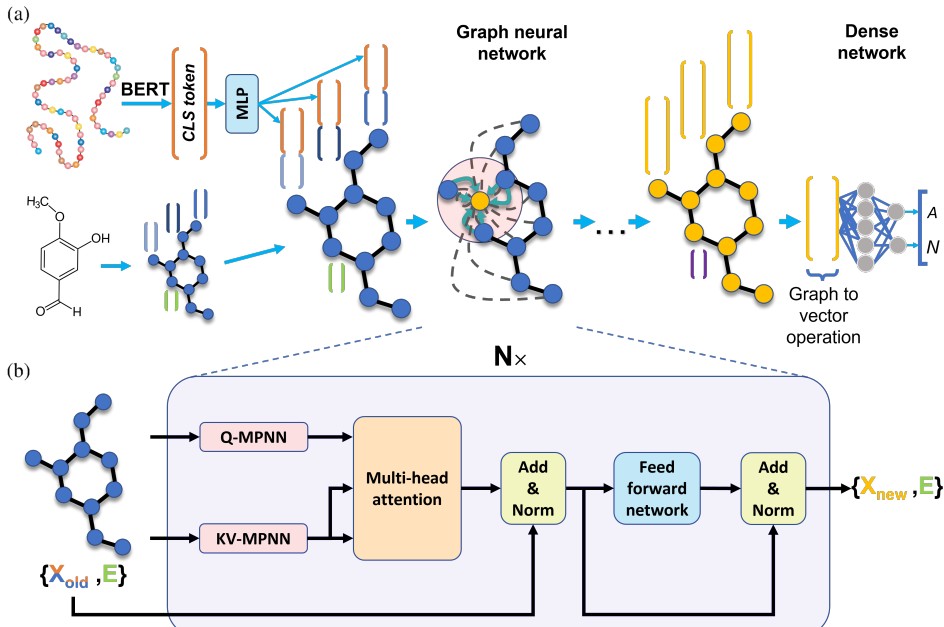

Figure 1: Model design. (a) Overview of the model. The input is a pair of protein sequence and molecular graph. The sequence is embedded using [CLS] token from protBERT (Elnaggar et al., 2021) and the resulting representation is concatenated to each node of the molecular graph. (b) Graph processing block. The node embeddings $X_{old}$ of the input graph are first locally updated using two identical message passing neural networks (Q-MPNN and KV-MPNN), and then pairwise interactions are modeled by the multi-head attention. This is followed by residual connections, layer normalization and a feed forward network (Vaswani et al., 2017) to obtain the updated node embeddings $X_{new}$. Graph to vector operation is an ECC layer (Simonovsky & Komodakis, 2017) followed by an attention pooling (Section A, equation (2)). Edge features $E$ are used in Q-MPNN and KV-MPNN and are reset at the end of the block.

followed by layer normalization (Ba et al., 2016) and feed forward network (Vaswani et al., 2017). A similar approach was taken in (Rong et al., 2020), but our methodologies differ. Rong et al. (2020) replace linear transformations in the multi-head attention with graph neural networks and thus have $3 * |heads| * |layers|$ different message passing steps. Instead, we use two GNNs to construct queries and keys/values, and the standard multi-head attention with linear transformations, so our model has $2 * N * |layers|$ steps, where $N$ is the number of graph processing blocks.

The final model simply consists of $N$ blocks followed by an ECC layer (Simonovsky & Komodakis, 2017) and an attention pooling as the graph to vector operation, and a dense layer for final output. We chose to keep the GNNs architecture simple since there is a subsequent processing. We use a dense layer followed by ReLU to construct messages from concatenated sender, receiver and edge features, sum aggregation and a GRU cell (Cho et al., 2014) to update nodes[1].

**Training.** To avoid overfitting we adopt similar strategy as in (Rong et al., 2020). During training, a random number of message passing steps $T$ is sampled from a truncated normal distribution $N(\mu, \sigma, a, b)$ in each iteration of gradient descent. We use the Adam optimizer (Kingma & Ba, 2015) with the learning rate schedule from (Vaswani et al., 2017) and in all experiments we train the model for 10 000 epochs. In the first 8000 epochs we train on graphs with up to 32 nodes and 64 edges, and then we switch to the full dataset during the remaining 2000 epochs. For evaluation and prediction, we set the number of message passing steps to $\mu$. In our experiments, we set $\mu = 6$, $\sigma = 1$ and the truncation interval between $a = 3$ and $b = 9$.

---

[1]Model source code is available here: https://github.com/MatejHl/Receptor2Odorant

## 4 DATA

Only a limited amount of organized and curated data on OR-molecule pairs is available (Cong et al., 2022; Sharma et al., 2021; Liu et al., 2011). To fill this gap, we have gathered and curated a new dataset of 46 700 unique OR-molecule pairs from the literature, tripling the size of the currently largest database of this kind (Cong et al., 2022). The data puts together experiments for 1237 unique sequences for 11 mammalian species with additional information on experimental procedure and varying data quality. See Supplementary material G for further details about the dataset.

Mammalian olfaction is sensitive to chirality and two enantiomers can have different smell perception. However, 21% of the gathered data are actually pairs of an OR and a mixture of enantiomers. To address this, we consider two modeling strategies for the mixtures of enantiomers: a "mixture" approach where we exploit the ability of our proposed model to work with mixtures and a "single" approach where a mixture of enantiomers is treated implicitly, by changing the chiral property of the nodes. See further details in sections B and E.1.

In addition, there are several biases in the data that need to be addressed during training. Label noise coming from high-throughput screening, class imbalance, and the combinatorial nature of the task. In response, we construct sample weights for each sub-problem and use a weighted binary cross entropy loss where the final weights are the product of the partial weights. Further details for weights construction are discussed in Supplementary material C.

### 4.1 TEST SET

Because of the varying data quality, special attention needs to be paid to the test set construction. In order to have high confidence in the results we select only pairs from the highest quality dose-response data for the test set. Unless otherwise stated, in each cross-validation run in the experiments, we randomly select 30% of the dose-response samples for the test set. We run 5 cross-validation runs for all experiments.

## 5 EXPERIMENTS

We performed a series of experiments on the newly gathered dataset of OR-molecule pairs. In 5.1 we compare our proposed architecture with other GNNs and perform ablation studies on the inductive biases. We explore the generalization of the model to unseen receptors and molecules in Section 5.2, and in 5.3 we compare our approach with the previous studies of OR-molecule interactions. We present additional ablations and experiments on two other datasets in Supplementary material E. Due to the imbalance nature of molecule-protein activations, we use Mathews correlation coefficient (MCC) as the main evaluation metric. We also report others for comparison with previous work.

### 5.1 MODEL

**GNN architecture**  We compare our model with 4 standard GNN models and we experiment with 2 additional modifications of our approach. We also compare our results with 2 state-of-the-art models on drug-target interaction prediction (DTI), namely HyperAttentionDTI (Zhao et al., 2021) and MolTrans (Huang et al., 2020), which based on Zhao et al. (2021) perform best on three different DTI datasets. We use ECC (Simonovsky & Komodakis, 2017), GGNN (Li et al., 2015), GAT (Veličković et al., 2018) and GIN (Xu et al., 2019) as standard GNN baselines. GAT is a particularly interesting baseline because it is a generalization of attention mechanism to an arbitrary graph. In the first ablation, we consider dropping the *non-bonded interactions* observation and we replace Q-MPNN, KV-MPNN and the multi-head attention in Fig. 1 (b) with a GAT layer (Transformer GAT in Tab. 1). This way the attention mechanism is kept, but we do not allow direct interactions between all pairs of atoms, and the model is restricted solely to the topology of the molecule. The second ablation is in the opposite direction. We drop the molecular topology and use only the multi-head attention by setting Q-MPNN and KV-MPNN to identity functions. Note that this reduces to the Transformer (Vaswani et al., 2017) encoder model.

As can be observed in the results in Tab. 1, our model outperforms the standard baselines and also proves to be the best compared to the DTI models. The ECC model failed to converge at all and its

Table 1: Summary of results for different architectures. *Transformer GAT* corresponds to replacing Q-MPNN, K-MPNN and the multi-head attention in Fig. 1 (b) with the GAT layer, and *Transformer* corresponds to setting Q-MPNN and K-MPNN to identity functions. All ablation models have 5 layers/blocks. Since baseline models cannot process mixtures, we use "single" approach to represent mixtures of enantiomers for all models except *Ours - mixture* (see Section B for details).

| Model | AveP | Precision | Recall | F-score | MCC |
|---|---|---|---|---|---|
| ECC | 0.307 (0.02) | 0.325 (0.02) | 0.527 (0.03) | 0.402 (0.03) | 0.187 (0.03) |
| GGNN | 0.676 (0.02) | 0.447 (0.02) | 0.833 (0.02) | 0.582 (0.02) | 0.455 (0.02) |
| GAT | 0.686 (0.03) | 0.466 (0.03) | 0.834 (0.03) | 0.597 (0.02) | 0.476 (0.02) |
| Transformer GAT | 0.704 (0.02) | 0.556 (0.01) | 0.771 (0.04) | 0.645 (0.01) | 0.536 (0.01) |
| GIN | 0.743 (0.02) | 0.641 (0.03) | 0.708 (0.04) | 0.672 (0.02) | 0.574 (0.02) |
| Transformer | 0.748 (0.04) | 0.654 (0.04) | 0.695 (0.05) | 0.671 (0.01) | 0.576 (0.01) |
| MolTrans | 0.638 (0.07) | 0.402 (0.05) | 0.822 (0.03) | 0.556 (0.05) | 0.476 (0.04) |
| HyperAttentionDTI | 0.737 (0.02) | 0.609 (0.03) | 0.773 (0.02) | 0.681 (0.02) | 0.584 (0.02) |
| Ours - single | 0.765 (0.02) | 0.665 (0.01) | 0.711 (0.02) | 0.687 (0.02) | **0.595** (0.02) |
| Ours - mixture | 0.780 (0.01) | 0.689 (0.02) | 0.698 (0.04) | 0.693 (0.02) | **0.605** (0.02) |

Table 2: Summary of results for strategies to combine sequence and molecule inputs. Mixtures of enantiomers are treated as a single graph.

| Mixing | AveP | Precision | Recall | F-score | MCC |
|---|---|---|---|---|---|
| Concatenation | 0.687 (0.03) | 0.550 (0.05) | 0.729 (0.02) | 0.626 (0.03) | 0.509 (0.03) |
| Attention | 0.720 (0.02) | 0.604 (0.02) | 0.671 (0.03) | 0.635 (0.02) | 0.526 (0.02) |
| Ours - single | 0.765 (0.02) | 0.665 (0.01) | 0.711 (0.02) | 0.687 (0.02) | **0.595** (0.02) |

performance is poor compared to the others. The best from the baseline models is GIN with MCC 0.574, followed by GAT and GGNN. Compared to simple GAT, the performance increases in the first ablation when we combine the GAT layer with feed-forward network, normalization, and residual connections. Interestingly, ignoring the molecular topology and using Transformer yields better performance than any of the baseline GNNs or than Tranformer GAT. This is surprising because Transformer has no access to the topology of the molecule and this suggests that the multi-head attention component is crucial for performance. However, combining both the multi-head attention and GNN together works the best (our model), implying that both the molecular topology and pairwise interactions are needed. Comparing to the DTI models, MolTrans performs worse than most of the baseline models and ablations. Our approach surpasses it by a large margin of 0.119 (0.129 with the "mixture" approach). HyperAttentionDTI performs better than MolTrans and the baseline GNNs, but is still outperformed by our approach.

**Mixing** It is not evident at which point in the model the protein and molecule inputs should be mixed. We argue that the molecule and the receptor create a complex, and since they constantly interact with each other, the information should be mixed in an early stage of the processing. We test this hypothesis against a more common strategy (Du et al., 2022) of late mixing of protein and molecule information. We compare our model with a standard strategy of processing molecular information separately via a GNN and then mixing the molecular embedding with the protein representation. This breaks the *locality* and *receptor-molecule complex* observations. We consider two mixing strategies: concatenation and attention. Concatenation is the most standard one and in this case the molecule is processed without the protein information and the resulting embedding from the graph to vector operation is concatenated to the protein [CLS] token. This concatenated vector is then the input to a dense network. The second strategy is to use the resulting molecule embedding as keys/values in the multi-head attention and the [CLS] representation of the protein as a query. This is followed by layer normalization and feed-forward network like in the Transformer block. We use the proposed blocks in Fig. 1 (b) to generate molecular embedding.

The results for mixing ablation are summarized in Tab. 2. Despite the redundancy of the protein representation in the nodes, our approach outperforms other mixing strategies. This suggests that by

incorporating the *locality* and *receptor-molecule complex* inductive biases, the model can learn to exploit the dependencies between various molecular binding groups and the receptor.

## 5.2 GENERALIZATION

An important and perhaps the most interesting experiment we conduct is to test the generalization of the model. 43% of mammalian ORs have no identified active compound, and only 206 out of 385 human ORs have a known responsive molecule. Therefore, the prediction of active compounds for so-called orphan receptors (i.e. deorphanization task) remains a great challenge. Moreover, among 6389 described odorant molecules (Castro et al., 2022), 504 have been tested so far, and finally only 336 of them are known to induce an OR response. Thus, it is of major interest to identify new active molecules for ORs (i.e. expanding chemical space).

**Deorphanization.** Model generalization to unseen ORs is tested in two scenarios. In the first one (called *random* in Tab. 3), several ORs are randomly chosen to be exclusively included in the test set and all their occurrences are removed from the training set. This scenario appears more realistic because deorphanized ORs are found in each OR family. In the second, more complex scenario, entire clusters of receptors are selected for the test set. The ORs are clustered according to their sequence similarity, and then all dose-response pairs for all receptors in 9 randomly picked clusters are placed in the test set. In this complex case, the model has to extrapolate to proteins with dissimilar sequences. In both scenarios, we also experiment with keeping negative primary and secondary screening pairs for the selected receptors (*OR-keep* in Tab. 3).

**Expanding chemical space.** Identifying new active compounds for olfactory receptors remains challenging. To test the ability of the model to generalize to new molecules, we follow the same strategy as for "deorphanization" (*Molecules* in Tab. 3). We either randomly select molecules and put all their occurrences to the test set, or we choose several clusters based on Tanimoto coefficient (i.e. structural similarity). See Section D for the full description of the experimental setup.

**Generalization results.** The results of the generalization experiments are summarized in Tab. 3. In the first *random* scenario, mainly recall is affected compared to an i.i.d. split and precision undergoes minor changes by 2-5%. Keeping non-active molecules has small impact and leads to a more conservative model with higher precision and lower recall. It appears that the model generalizes poorly to entirely new family of receptors (Tab. 3, *cluster* task). This is due to the fact that subtle variations in the amino acids sequence of ORs can utterly alter its recognition profile (de March et al., 2018; Trimmer et al., 2019). Interestingly, even though overall MCC is low for the *cluster* scenario, the precision is still maintained at a relatively high level with a difference of 11-15% compared to the i.i.d. split. From the molecule perspective, the model can generalize well to new compounds that have similar distribution to the training set. As expected, the performance decreases on an entirely unseen group of molecules, yet the model still keeps the precision of 54.4%, only 14% lower than in the i.i.d. case. The recall on the other hand deteriorates by half. This suggests that the model is conservative and reluctant to predict many responsive pairs, which can be an advantage for *in vitro* experiments. Although some responsive pairs are missed as a result of lower recall, the predictions keep similar reliability across scenarios, and thus the number of laboratory hits would remain stable.

## 5.3 COMPARISON WITH OR-MOLECULE MODELS

In Tab. 4, we compare our work with Kowalewski & Ray (2020), Cong et al. (2022), and Gupta et al. (2021). Overall, when generality is taken into account, our model outperforms previous approaches. In particular, Kowalewski & Ray (2020) perform slightly better in AUROC than others, but this approach fits a separate SVM model for each OR and can only be generalized to ORs that already have at least 3 known active compounds. This is a limitation for most of ORs. On the other hand, Cong et al. (2022) took a more general approach by fitting a single model to data for 10 molecules (4 targets and 6 similar compounds). It is applicable to a variety of ORs, but only for compounds structurally close to the 4 targets. Even though Cong et al. (2022) consider an easier setting with a limited number of molecules, our approach outperforms it by a large margin. Finally, our model and the BiLSTM model of Gupta et al. (2021) consider any given molecule-receptor pair. Our approach outperforms Gupta et al. (2021) in all the metrics and more than twice the precision.

Table 3: Summary of results for a generalization test set in case that a cluster of molecules/sequences is not present in the training set (*Cluster*) and randomly chosen molecules/sequences are not present in the training set (*Random*). *OR - keep* corresponds to a case where primary and secondary screening non-active molecules are kept in the training set. Mixtures of enantiomers are treated as a "mixture".

| | Split | Num data | AveP | Precision | Recall | F-score | MCC |
|---|---|---|---|---|---|---|---|
| | i.i.d. | 1565.0 | 0.780 (0.01) | 0.689 (0.02) | 0.698 (0.04) | 0.693 (0.02) | 0.605 (0.02) |
| Cluster | Molecule | 1320.4 | 0.580 (0.08) | 0.544 (0.07) | 0.342 (0.06) | 0.418 (0.06) | **0.334** (0.07) |
| | OR | 1062.0 | 0.558 (0.14) | 0.535 (0.12) | 0.132 (0.04) | 0.203 (0.04) | **0.088** (0.06) |
| | OR - keep | 1062.0 | 0.625 (0.03) | 0.576 (0.06) | 0.095 (0.03) | 0.161 (0.05) | **0.091** (0.09) |
| Random | Molecule | 1056.2 | 0.729 (0.08) | 0.657 (0.11) | 0.629 (0.04) | 0.638 (0.06) | **0.533** (0.07) |
| | OR | 1217.8 | 0.684 (0.10) | 0.636 (0.07) | 0.491 (0.11) | 0.552 (0.09) | **0.417** (0.10) |
| | OR - keep | 1217.8 | 0.710 (0.09) | 0.670 (0.06) | 0.470 (0.13) | 0.548 (0.11) | **0.430** (0.10) |

Table 4: Comparison with previous studies on OR-molecule activation prediction. *ORs* and *Mols* are average numbers of unique receptors and molecules, respectively, in the test set.

| Model | ORs | Mols | TNR | AUROC | Precision | Recall | F-score |
|---|---|---|---|---|---|---|---|
| SVM[a] | 32 | n/a | 0.80 (0.06) | **0.88** (0.07) | n/a | **0.77** (0.10) | n/a |
| RF[b] | 80 | 4 | **0.91** | 0.74 | 0.52 | 0.53 | 0.53 |
| BiLSTM[c] | n/a | n/a | n/a | 0.77 | 0.34 | 0.71 | 0.46 |
| Ours - mixture | **205** | **237** | **0.91** (0.01) | 0.79 (0.03) | **0.69** (0.02) | 0.70 (0.04) | **0.69** (0.02) |

[a]Kowalewski & Ray (2020)
[b]Cong et al. (2022)
[c]Gupta et al. (2021), results taken from supplementary Fig. 3.

## 6 AGREEMENT WITH ODOR PERCEPTION

The combinatorial nature of olfaction was first hypothesized in 1973 by Polak (1973) and further confirmed by the discovery of ORs by Buck & Axel (1991). Malnic et al. (1999) demonstrated that different odorants are recognized by different subsets of ORs. Here, we refer to a subset of 385 human ORs activated by a given odorant as the combinatorial code, and in this section we asses a biological plausibility of our approach by analyzing consistency between the combinatorial code predicted by our model and odor perception of molecules (i.e. the odor families).

We evaluate the concordance between our model and the olfactory perception as follows: given a large dataset of known odorants (Castro et al., 2022), we construct an abstract smell characteristic of a molecule by taking an embedding of a GNN-based odor prediction model (Sanchez-Lengeling et al., 2019). This model was recently shown to outperform a median panelist in odor description task, and its embedding can capture perceptual similarity of molecules (Lee et al., 2022). The odorants are clustered based on their odor embedding to abstract odor families. For each family, we define an agreement score as the $\alpha$-quantile of the distribution of median $l_1$ distances between the combinatorial codes within the family[2]. In other words, let $A_i^k, A_j^k \in \{0, 1\}^{385}$ be the combinatorial codes for molecules $i, j$ in family $k$. Then the agreement score $Q_k$ for family $k$ is given by

$$Q_k(\alpha) = \inf_q \{\forall i \in k : P(\hat{m}\{\|A_i^k - A_j^k\|_1\} \le q) > \alpha\} \tag{1}$$

where $\hat{m}$ denotes the median through $j \in k$. When considering $\alpha$ percent of the most consistent codes in family $k$, $Q_k(\alpha)$ represents the largest deviation of OR responses in that family. We report several values of $\alpha$ due to the known ambiguity of odor descriptors which depend on several cultural and personal factors of the panelists such as varying verbal expressions, current state and previous experience, or are even affected by OR mutations (Kaeppler & Mueller, 2013).

Fig. 2 reports the distribution of $Q_k(\alpha)$ for all abstract odor families and several levels $\alpha$. Most of the families have a consistent combinatorial code as illustrated by the density of $Q_k$ (Fig. 2), highlighting

[2]There are 205 abstract odor families and 385 human ORs.

(a) 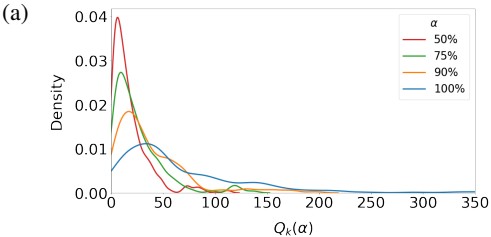 (b) 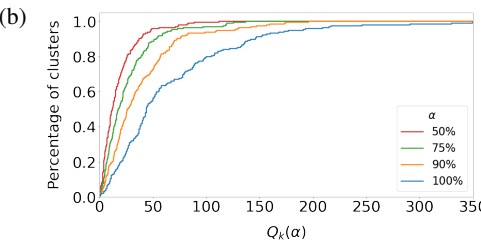

Figure 2: (a) Density and (b) cumulative distribution of agreement scores $Q_k(\alpha)$ for several levels $\alpha$.

an agreement between the results of the two independent models: odor prediction and OR activation. For $\alpha = 75\%$, more than $60\%$ of all families differ by less than 25 ORs and $84\%$ by less than 50. Fig. 5 provides examples of families with different $\alpha$-quantile values. Although the odor prediction model outperforms a median panelist, it is still prone to error. This may influence the quality of the clustering and consequently lead to outliers in the families, as illustrated in Fig. 5 (b).

A previous *in vitro* study on 125 odorants suggested that molecules recognized by a subset of ORs share the same odor quality (Nara et al., 2011). The consistency of activation patterns observed in the odor families allows us to generalize this experimental observation to all known odorants. Nara et al. (2011) found that most ORs are narrowly tuned with the exception of few broadly tuned ORs. Our predictions are consistent with this observation, as the results conclude that the vast majority of ORs have a very narrow recognition spectrum (Fig. 4). In addition, our model is able to identify specialized receptors for some specific odor families (e.g. Fig. 6). In agreement with Nara et al. (2011), odor is generally encoded in a combination of multiple active receptors and the size of the code varies among odorants.

## 7 CONCLUSION

In this study, we transfer the biological problem of protein-molecule interaction to a graph-level binary classification. We combine [CLS] token from the protein language model with graph neural networks and multi-head attention to predict active compounds for ORs. We design a tailored GNN architecture by incorporating inductive biases that are observed in receptor-ligand interactions and we demonstrate that this architecture outperforms other baselines and previous works. Our results show that [CLS] token contains valuable information to characterize proteins and at the same time lowers computational and memory demands. In addition, by taking into account non-bonded interactions, the model is suited to deal with mixtures of compounds.

Although [CLS] token reduces computational costs, it comes with the limitation that one cannot infer the importance of individual amino acids. In the future, we plan to explore the possibility to use MSA-based protein language models to explicitly model amino acids. This may lead to better interpretability with the drawback of higher computational demands. While our predictions are in good agreement with experimental results, our model does not take into account the influence of the concentration on either the perceived odor or the pattern of activated ORs. Recent experiments have shown that inhibition of ORs by odorants is also part of the combinatorial code of odors and, more specifically, that the interaction between odorants in a mixture modulates the activation of ORs relative to that of isolated molecules (Kurian et al., 2021). However, data on OR inhibition remain scarce and do not allow training a model predicting molecular inhibition.

To date, 43% of the ORs are still orphans. The prediction of the combinatorial code for all ORs and odorant molecules opens the way to the generalization of experimental results obtained on subsets of ORs or a limited number of odorants. By linking activity patterns to olfactory perception, we confirm that odorants are recognized by a unique subset of ORs and that these subsets are specific to an olfactory characteristic. Finally, we expect this work to have an impact beyond the olfaction research. ORs are also expressed in non-nasal tissues and are involved in the regulation of different metabolic functions or found in cancerous cells. Thus, they are becoming a promising drug targets, and matching ligand to specific ORs is of major importance.

ACKNOWLEDGMENTS

We thank Jana Bocková and Jody Pacalon for helpful feedback on drafts of this paper and for insightful discussions on the activation mechanism of ORs. This work was funded by the French National Research Agency (ANR) under reference number ANR-19-CE07-0044 (PhD fellowship to Matej Hladiš), by the Fondation Roudnitska under the aegis of Fondation de France (PhD fellowship to Maxence Lalis), by GIRACT, and it has also received financial support from the CNRS through the MITI interdisciplinary program. This work was granted access to the HPC resources of IDRIS under the allocation 2022-[AD010713580] made by GENCI and it was also supported by the French government, through the UCAJEDI Investments in the Future project managed by the ANR under reference number ANR-15-IDEX-01. The authors are grateful to the OPAL infrastructure from Université Côte d'Azur and the Université Côte d'Azur's Center for High-Performance Computing for providing resources and support.

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

## A  MODEL DETAILS

In all our experiments, we consider node and bond features in Tab. 5. Molecular graphs were constructed using RDkit (Landrum, 2020) from SMILES obtained from PubChem (Kim et al., 2020). Atomic number, chiral tag and hybridization were embedded in $\mathbb{R}^{72}$, and Bond type and Stereo type in $\mathbb{R}^{36}$.

Table 5: Features of the molecular graph. Features were treated as categorical variables except for *formal charge*, *number of implicit Hs*, *explicit valence* and *mass* which are continuous variables.

| Atom features | Bond features |
| --- | --- |
| Atomic number | Bond type |
| Chiral tag | Stereo type |
| Hybridization | Is aromatic |
| Formal charge | |
| Num. of implicit Hs | |
| Explicit valence | |
| Mass | |
| Is aromatic | |

The attention pooling layer in graph to vector operation for graph $g$ is defined as

$$\alpha = softmax(Xw) \qquad x' = \sum_{i=1}^{n} \alpha_i X_i \tag{2}$$

where $X \in \mathbb{R}^{n \times d}$ is a matrix of node embeddings and $w \in \mathbb{R}^{d \times 1}$ are trainable parameters.

### A.1  TRAINING DETAILS

Dropout of $0.1$ was applied in each multi-head attention layer and $0.5$ just before the output layer. Padding was set to 32 nodes and 64 edges in the first 8000 epochs and 128 and 256, respectively, in the remaining 2000 epochs. The initial learning rate was set to 0.001 and we use 6000 warm-up steps for the scheduler. The model was implemented in JAX (Bradbury et al., 2018) and FLAX (Heek et al., 2020) and trained on Nvidia V100 SXM2 32GB or Nvidia A100 SXM4 80GB GPUs.

## B  ENANTIOMERS

The olfactory system of primates is sensitive to the chirality of molecules: they can discriminate between enantiomers[3] are mirror isomers that are not superimposable chemical structures. They have the same atomic composition and they share identical physical and chemical properties in an achiral environment. However, they will have different properties when interacting with a chiral environment (for example, polarized light or the binding site of a protein). Two enantiomers can have a completely different odor, for example *R*-carvone and *S*-carvone which smell like caraway and mint, respectively. At the molecular level, these two molecules are recognized by distinct OR subsets. The specific selectivity of ORs for a given enantiomer is at the origin of the differences in perception of chiral molecules. Even if chirality has a high influence on ORs recognition, the distinction between enantiomers is often overlooked in laboratory experiments. As a result, 21% of our gathered data are actually pairs of an OR and a mixture of two (or more) enantiomers. Thus, it appears crucial to inject molecular chirality into our model.

Molecular graphs of enantiomers are identical and the only difference is in their chiral centers. These may have clockwise (CW) or counter-clockwise (CCW) chirality and one way of distinguishing them is to put *chiral tag* property of one chiral center to CW and CCW for the other. Since the molecular graphs are identical, a mixture of enantiomers can be modeled in two ways: as a single graph where chiral centers have "unknown" chirality or a mixture of independent molecules. In the latter case, we can exploit the fact that combining GNN with the multi-head attention is capable

---

[3]With a slight abuse of notation, we refer to both enantiomers and diastereoisomers as *enantiomers*.

Table 6: Screening quality. Values in each row correspond to the conditional probability that the true response is positive/negative (i.e. the outcome of the dose-response measurement) given the result from the primary and secondary screening. These probabilities were estimated based on the data from Mainland et al. (2015). $EC_{50}$ denotes the true label from the dose-response measurements.

|  |  | Primary | | Secondary | |
| --- | --- | --- | --- | --- | --- |
|  |  | pos. | neg. | pos. | neg. |
| $EC_{50}$ | pos. | 0.40 | 0.31 | 0.72 | 0.23 |
|  | neg. | 0.60 | 0.69 | 0.28 | 0.77 |

of modeling mixtures. Thus, a mixture of enantiomers can be treated as a graph with $2^{|chiral\ centers|}$ identical disconnected components that differ only in the *chiral tag* property of chiral centers.

We test both "single" and "mixture" approaches in Section E.1. Our experiments indicate that the two strategies have similar performance on OR-molecule pairs (Tab. 7). Interestingly, when evaluated on a limited set of complex mixtures (i.e. pairs of OR and mixture containing more than two topologically different compounds), the performance of the model trained in the "mixture" approach is higher even though the only mixtures seen during training were the mixtures of enantiomers (Tab. 8). Note that the data on mixtures are limited and further experiments are needed.

## C  SAMPLE WEIGHTS

**Data quality.**  Conducting an experiment with high confidence on responsiveness is costly and time-consuming. The standard procedure for testing molecule-receptor pairs is to first perform a noisy primary screening where many different pairs are tested with just one injection for a single concentration. Then, promising pairs from the primary screening are further tested at several concentrations in a secondary screening. Finally, the pairs that are considered responsive from the secondary screening are tested in the most precise tertiary screening in multiple concentrations and multiple injections per concentration (i.e. dose-response curve). If neglected, varying sample quality could lead to fitting corrupted labels and impairing the model performance and generalization. Therefore, during training we employ a simple strategy to counteract this label noise. For each type of screening, we estimate probability $P_{c,\hat{c}}^{type}$ that the observed label $\hat{c}$ corresponds to the true label $c$. By treating a dose-response label as true we estimate $\hat{P}_{c,\hat{c}}^{type}$ based on (Mainland et al., 2015), where several pairs were tested in all three types of experiments (Tab. 6). Then we use the diagonal elements as data quality weights for a given type and a given class $y$

$$w_{quality}(y, type) = \hat{P}_{y,y}^{type} \tag{3}$$

We set the weight for dose-response samples to 1. In the future, a more elaborated label noise treatment could be used (see (Song et al., 2022) for a non-exhaustive list of label noise strategies).

**Class imbalance.**  Naturally, most of the molecule-receptor pairs are non-responsive and only $6.2\%$ of the samples in our data are responsive. To account for this class imbalance, we use standard class weights based on the imbalance ratio

$$w_{class}(y) = \begin{cases} \frac{|non-responsive|}{|responsive|} & \text{if } y = 1 \\ 1 & \text{if } y = 0 \end{cases} \tag{4}$$

**Pair imbalance.**  On top of the class imbalance, some olfactory receptors have been tested with many molecules, whereas others are yet to be explored and were tested only on a few specific compounds (Fig. 7). We refer to this type of imbalance as *pair imbalance* and to avoid the bias resulting from the uneven distribution of experiments, we adopt a heuristic weighting scheme to down-weight extensively tested receptors and up-weight less-known ones. For each molecule-receptor pair, we set pair imbalance weight as the logarithm of the harmonic mean

$$w_{pair}(m, r) = ln\left(1 + \frac{K}{2}\left(\frac{1}{|M(r)|} + \frac{1}{|OR(m)|}\right)\right) \tag{5}$$

where $|M(r)|$ is the number of molecules tested for a given receptor $r$, $|OR(m)|$ is the number of receptors tested for a given molecule $m$ and $K$ is a constant. In the experiments, we set $K = 100$.

The final sample weight is then given by

$$w(m, r, y, type) = w_{quality}(y, type)w_{class}(y)w_{pair}(m, r) \qquad (6)$$

## D   EXPERIMENTAL DETAILS

### D.1   GENERALIZATION

**Deorphanization.**   To test the generalization to unseen receptors we consider two scenarios. The first one (referred to as *random* in Tab. 3), where the model predicts active compounds for unseen receptors chosen randomly, and the second, more complex scenario, where entire groups of similar receptors are unseen during the training (*cluster* in Tab. 3).

We test the *random* scenario by constructing a test set for each cross-validation run in the following way. From pairs tested in dose-response experiments, we filter receptors with at least 3 known active compounds. Then from these filtered receptors we randomly choose 35% and place all dose-response pairs containing these into the test set. We then discard remaining primary and secondary screening pairs containing the selected receptors from the training set (OR in Tab. 3) or we discard only the responsive pairs containing these receptors (OR - keep in Tab. 3). The reasoning behind keeping negative examples is that, in reality, we have access to non-responsive molecules from primary screenings for all human olfactory receptors. Following this procedure, the model has never seen the selected receptors and it needs to extrapolate their molecule-receptor interaction from the remaining proteins.

Test set construction in the *cluster* scenario is similar to the *random* case, but we do not choose receptors for the test set uniformly. Instead, we cluster receptors based on their sequence similarity and then place dose-response pairs for all receptors from 9 randomly picked clusters into the test set. We then follow the same procedure of discarding either all or only the responsive primary and secondary screening pairs containing these receptors from the training set. This scenario is more demanding than the first setting because the model would need to extrapolate to entirely dissimilar proteins than the ones in the training set.

**Expanding chemical space.**   Extrapolation to the unseen molecules is tested by an identical procedure as the deorphanization. The only differences are that we select molecules instead of receptors (25% molecules with at least 2 responsive pairs in *random*) and in the *cluster* scenario we cluster molecules based on their Tanimoto coefficient (i.e. structural similarity). We take compounds from 6 randomly selected clusters. In both scenarios, we then discard all other pairs containing the selected compounds from the training set.

## E   ADDITIONAL EXPERIMENTS

### E.1   DATA ABLATIONS

**Enantiomers.**   As we argued previously, olfaction is sensitive to the chirality and modeling enantiomers poses a great challenge. Here, we test the two approaches to incorporate mixtures of enantiomers and their performance is summarized in Tab. 7. We report two settings where we either test the model on single molecules only (i.e. discarding mixtures of enantiomers from the test set) or on the full test set. We also train a separate model on the training set with mixtures of enantiomers discarded. As can be seen in the results, modeling mixtures of enantiomers explicitly as a mix of independent molecules slightly outperforms the implicit "single" approach. In both cases keeping mixtures of enantiomers is beneficial for modeling single molecules.

**Complex mixtures.**   Several lab experiments have been performed to test a receptor with a mixture of topologically different compounds. In Tab. 8, we report the performance of our model on a limited dataset of complex mixtures. In particular, we investigate impact of the two strategies for treating mixtures of enantiomers on the performance on complex mixtures. Although the two approaches

Table 7: Summary of results for strategies to treat mixtures of enantiomers. Mixtures of enantiomers are either modeled as a mixture of distinct molecules (*mixture*) or as a single graph with chiral centers having "unknown" chirality (*single*). *Discard* corresponds to the model trained only on data without the mixtures. *Discard mix of isomers* denotes a test set without the mixtures and *full test set* is a test set containing all pairs, including mixtures of enantiomers.

| | Isomers | Num data | AveP | Precision | Recall | F-score | MCC |
|---|---|---|---|---|---|---|---|
| Discard mix of isomers | Discard | 1425.0 | 0.773 (0.01) | 0.666 (0.03) | 0.689 (0.03) | 0.677 (0.02) | 0.589 (0.03) |
| | Single | 1425.0 | 0.759 (0.02) | 0.656 (0.01) | 0.715 (0.03) | 0.685 (0.02) | 0.597 (0.03) |
| | Mixture | 1425.0 | 0.778 (0.01) | 0.681 (0.02) | 0.704 (0.04) | 0.691 (0.02) | **0.608** (0.02) |
| Full test set | Single | 1565.0 | 0.765 (0.02) | 0.665 (0.01) | 0.711 (0.02) | 0.687 (0.02) | 0.595 (0.02) |
| | Mixture | 1565.0 | 0.780 (0.01) | 0.689 (0.02) | 0.698 (0.04) | 0.693 (0.02) | **0.605** (0.02) |

Table 8: Summary of results for a small dataset of complex mixture-receptor pairs for dose-response pairs only ($EC_{50}$) and all available data (*All*).

| | Isomers | Num data | AveP | Precision | Recall | F-score | MCC |
|---|---|---|---|---|---|---|---|
| $EC_{50}$ | Single | 37.0 | 0.938 (0.05) | 0.895 (0.04) | 0.836 (0.02) | 0.864 (0.01) | 0.494 (0.10) |
| | Mixture | 37.0 | 0.994 (0.01) | 0.915 (0.07) | 0.836 (0.07) | 0.869 (0.02) | **0.538** (0.10) |
| All | Single | 112.0 | 0.799 (0.08) | 0.716 (0.08) | 0.774 (0.03) | 0.741 (0.03) | 0.638 (0.05) |
| | Mixture | 112.0 | 0.940 (0.04) | 0.789 (0.10) | 0.755 (0.06) | 0.768 (0.04) | **0.684** (0.07) |

have identical training sets, the results suggest that the model trained with the "mixture" strategy outperforms the "single" approach.

**Data quality weights.** Label noise originating from different quality of experiments can bias the model and negatively impact its generalization. To test the label noise treatment, we experiment with discarding primary screening data (the most noisy ones in Tab. 6), with weights presented in Section C, and with ignoring the label noise (i.e. a naive approach). Note that we use only dose-response data for test sets and these are assumed to have no noise.

According to the results in Tab. 9, neglecting the label noise lowers the performance. It can be observed that when using weights and when discarding the primary screening data the results are on par. This is due to the fact that most of the corrupted labels are coming from the primary screening (Teb. 6) and so discarding these cures a large portion of the label noise. Nevertheless, we hypothesize that keeping the primary screening data while taking the noise into account could be beneficial for model generalization. Primary screening experiments have relatively low cost and can cover a large number of different pairs. Thus, the model can have access to a large sample of the data distribution.

Table 9: Summary of results for data quality weights. *Naive* corresponds to equal weights. Mixtures of enantiomers are treated as a "mixture".

| Data quality | AveP | Precision | Recall | F-score | MCC |
|---|---|---|---|---|---|
| Discard primary | 0.754 (0.02) | 0.667 (0.03) | 0.724 (0.04) | 0.694 (0.01) | 0.603 (0.02) |
| Naive | 0.740 (0.04) | 0.631 (0.05) | 0.715 (0.02) | 0.669 (0.03) | 0.570 (0.04) |
| Weight | 0.780 (0.01) | 0.689 (0.02) | 0.698 (0.04) | 0.693 (0.02) | **0.605** (0.02) |

### E.2 PERFORMANCE ON OTHER DATASETS

In addition to OR-molecule data, we conduct experiments on two drug-target interaction prediction datasets, namely KIBA (Tang et al., 2014) and DAVIS (Davis et al., 2011), and we compare performance of our approach with two state-of-the-art drug-target interaction models HyperAttentionDTI (Zhao et al., 2021) and MolTrans (Huang et al., 2020), which based on Zhao et al. (2021) perform best on the two datasets. Both KIBA and DAVIS record kinase inhibition binding affinities measured in laboratory assays. We follow data preprocessing and binarization scheme from (Zhao et al., 2021).

Table 10: Statistics of DAVIS (Davis et al., 2011), KIBA (Tang et al., 2014) and M2OR (ours) datasets. *Seqs.* and *Mols.* respectively refer to unique number of sequences and moleucles in the datasets. *Orphans* is a number of sequences without any known active compound and *% mols. tested per sequence* and *Ligands per sequence* are, respectively, average portion of molecules tested on a given sequence and average number of known active compounds for a sequence.

| Dataset | Num. pairs | Seqs. | Mols. | Orphans | % mols. tested per sequence | Ligands per sequence |
|---------|-----------|-------|-------|---------|------------------------------|----------------------|
| DAVIS | 25 772 | 379 | 68 | 0 | 100.0% | 19.3 |
| KIBA | 116 350 | 225 | 2068 | 5 | 25.0% | 98.5 |
| M2OR | 46 717 | 1237 | 596 | 527 | 6.3% | 2.3 |

Figure 3: Distributions of the number of laboratory experiments: (a) per molecule and (b) per sequence for DAVIS, (c) per molecule and (d) per sequence for KIBA. Note that DAVIS dataset contains results on all possible pairs of 68 molecules and 379 proteins.

In Tab. 10, Fig. 3 and Fig. 7 a comparison between distributions of the two datasets and of our data (M2OR in Tab. 10) is reported. As can be observed by shapes of the gray areas in Fig. 3 and Fig. 7, OR data have a more scarce distribution of tests per sequence and per molecule and a more imbalanced class distribution than what can be observed in KIBA or DAVIS.

Table 11: Performance of HyperAttentionDTI (Zhao et al., 2021), MolTrans (Huang et al., 2020) and our model on KIBA (Tang et al., 2014), DAVIS (Davis et al., 2011) and M2OR (ours) data. Results for MolTrans and HyperAttentionDTI on KIBA and DAVIS are taken from (Zhao et al., 2021).

| | Model | AveP | Precision | Recall | MCC |
|---|-------|------|-----------|--------|-----|
| **DAVIS** | MolTrans | 0.784 (0.002) | **0.782** (0.003) | 0.617 (0.004) | n/a |
| | HyperAttentionDTI | **0.839** (0.002) | 0.754 (0.002) | **0.780** (0.001) | n/a |
| | Ours - mixture | 0.744 (0.015) | 0.699 (0.013) | 0.666 (0.016) | 0.564 (0.011) |
| **KIBA** | MolTrans | 0.708 (0.003) | **0.710** (0.003) | 0.645 (0.003) | n/a |
| | HyperAttentionDTI | **0.814** (0.002) | 0.689 (0.003) | **0.798** (0.002) | n/a |
| | Ours - mixture | 0.704 (0.010) | 0.610 (0.014) | 0.683 (0.005) | 0.555 (0.012) |
| **M2OR** | MolTrans | 0.638 (0.066) | 0.402 (0.053) | **0.822** (0.027) | 0.476 (0.042) |
| | HyperAttentionDTI | 0.737 (0.015) | 0.609 (0.028) | 0.773 (0.020) | 0.584 (0.022) |
| | Ours - mixture | **0.780** (0.012) | **0.689** (0.016) | 0.698 (0.042) | **0.605** (0.017) |

The performance on all three datasets is summarized in Tab. 11. Overall, on DAVIS HyperAttentionDTI and MolTrans have higher AveP value than our model, and on KIBA HyperAttentionDTI performs best and our approach is on par with MolTrans. Our model outperforms other approaches on the OR-molecule dataset.

# F ANALYSIS OF THE COMBINATORIAL CODE

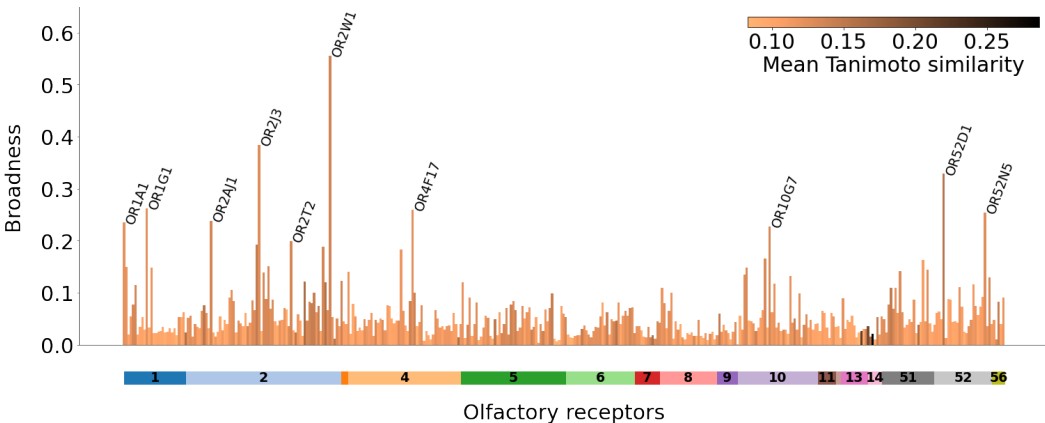

Figure 4: Distribution of the estimated broadness of human ORs. Broadness for each OR is defined as a portion of its predicted active compounds out of all known odorant molecules as given by the pyrfume database (Castro et al., 2022). The olfactory receptors are grouped by receptor families (Olender et al., 2008) and colored by mean Tanimoto similarity between their active compounds.

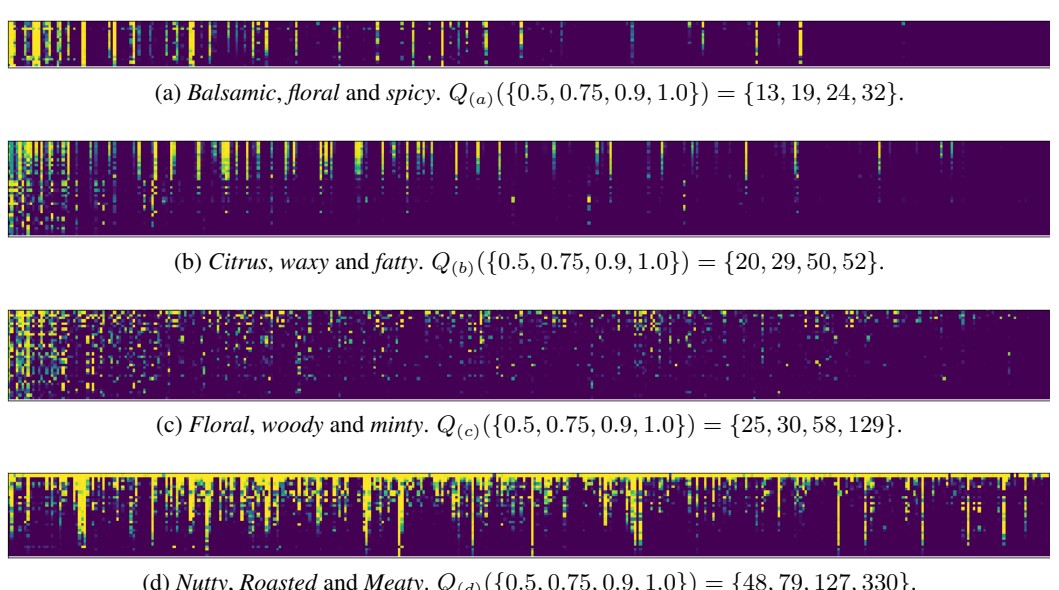

(a) *Balsamic*, *floral* and *spicy*. $Q_{(a)}(\{0.5, 0.75, 0.9, 1.0\}) = \{13, 19, 24, 32\}$.

(b) *Citrus*, *waxy* and *fatty*. $Q_{(b)}(\{0.5, 0.75, 0.9, 1.0\}) = \{20, 29, 50, 52\}$.

(c) *Floral*, *woody* and *minty*. $Q_{(c)}(\{0.5, 0.75, 0.9, 1.0\}) = \{25, 30, 58, 129\}$.

(d) *Nutty*, *Roasted* and *Meaty*. $Q_{(d)}(\{0.5, 0.75, 0.9, 1.0\}) = \{48, 79, 127, 330\}$.

Figure 5: Examples of the predicted combinatorial codes for 4 abstract odor families and 385 human ORs[4]. Each row is an odorant molecule and each column is an olfactory receptor. Each molecule-OR pair is colored by the probability of activation ranging from dark blue for no activation to yellow for an active pair. The ORs are sorted in the figures in descending order from the left to right by their predicted broadness. Captions under the figures correspond to the most frequent descriptors in a given abstract odor family, ordered by frequency.

---

[4]Figures for all families available here: `https://github.com/MatejHl/Receptor2Odorant`.

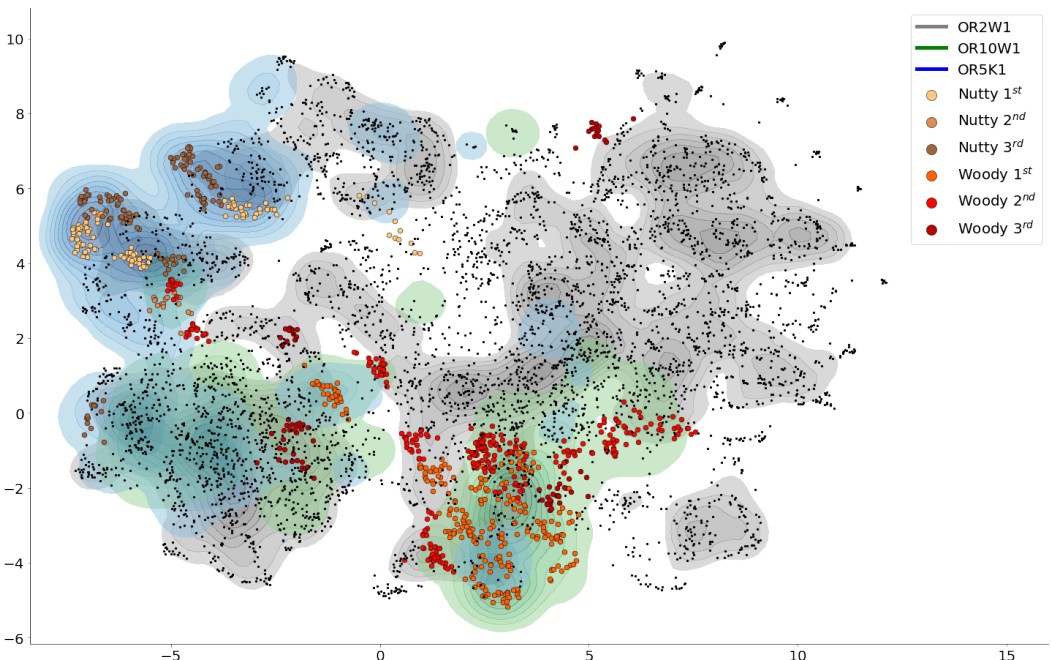

Figure 6: UMAP (McInnes et al., 2018) projection of the odor embedding with estimated densities of active compounds. Each point is an embedded odorant from pyrfume database (Castro et al., 2022) and highlighted points are molecules that belong to the abstract odor families with the given smell descriptor being 1. the most occurring one, 2. the second most occurring, and 3. the third most occurring in the family. The contour regions are estimated densities of active compounds for a given receptor. We identify specific receptors OR10W1 (green) and OR5K1 (blue). OR10W1 is activated by molecules with the *woody* descriptor whereas OR5K1 by *nutty* compounds. OR2W1 (grey) is an example of a broad receptor activated by odorants with all kinds of different descriptors.

## G  DATASET

Our newly gathered dataset of OR-molecule pairs[5], called M2OR for Molecule to Olfactory Receptor, currently contains 46 700 unique pairs that were collected from 31 scientific papers (Malnic et al., 1999; Wetzel et al., 1999; Kajiya et al., 2001; Spehr et al., 2003; Saito et al., 2004; Araneda et al., 2004; Sanz et al., 2005; Neuhaus et al., 2006; Jacquier et al., 2006; Schmiedeberg et al., 2007; Keller et al., 2007; Fujita et al., 2007; Saito et al., 2009; Repicky & Luetje, 2009; Shirasu et al., 2014; Mainland et al., 2015; Gonzalez-Kristeller et al., 2015; Geithe et al., 2015; 2016; 2017; Sato-Akuhara et al., 2016; Noe et al., 2016; 2017; Jones et al., 2019; Yasi et al., 2019; Frey et al., 2020; Jabeen et al., 2021; Marcinek et al., 2021; Haag et al., 2021; 2022; Cong et al., 2022). It combines 596 different molecules with 1237 unique sequences and in total consists of 68 837 experiments. Wild type, variant and mutant receptors are assembled across 11 different mammal species. For each experiment, several types of information are available:

- *Receptor*: Species, Mutation, Gene Name, Uniprot ID, Sequence. When available, primary sequence or Uniprot ID is obtained from the publication or else Uniprot ID is inferred from the gene name. We keep original gene name as was stated in the article. For mutants, Uniprot ID/Sequence corresponds to the reference sequence and the mutation is annotated in the dedicated column. Then mutated sequence should be deduced by combining the reference sequence and the mutation.

- *Molecule*: Name, CID, CAS, InChI Key, canonicalSMILES, Mixture. Compounds are identified by InChI Key based on the Pubchem database (Kim et al., 2020). Distinction between mono-molecular compounds, mixture, and sum of isomers is made, and it is derived

---

[5]Data available here: `https://github.com/chemosim-lab/M2OR`

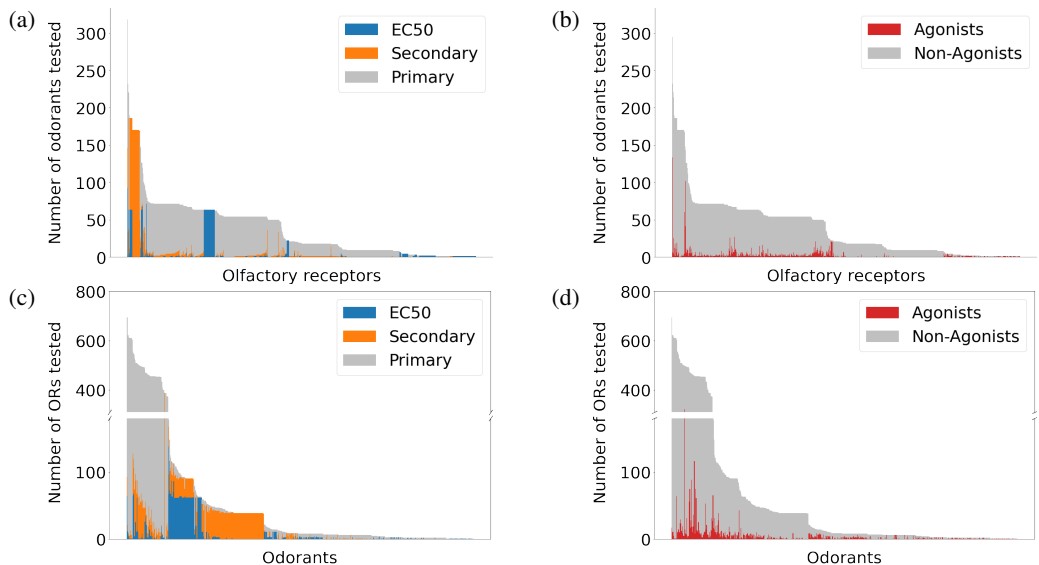

Figure 7: Distributions of the number of lab experiments in the M2OR dataset: (a-b) per receptor, (c-d) per molecule. Sever imbalance in the quality, number of experiments, and class distribution can be observed.

using *EnumerateStereoisomers* function form RDKit (Landrum, 2020). When no InChI Key is available, canonicalSMILES is inferred from the chemical structure.

- *Response*: Value, Unit, Value_Screen, Unit_Screen, Responsive, Nbr_measurements. Value corresponds to the experiment's raw response if available and Value_Screen is a tested concentration. If a sample is non-responsive in a dose-response measurement, Value_Screen is set to the maximum tested concentration. Nbr_measurements corresponds to the number of injections for a given experiment. Responsiveness is encoded as 1 for agonists and 0 for non-agonists.

- *Bioassay*: Type, Cell_line, Delivery, Assay, G-protein, Co_transfection, Assay System, Tag. Information about the type of assay done, the cell line used, the delivery method of the odorant and about the system such as the presence of G protein, tags or co-transfection of other proteins.

- *Reference*: Reference, DOI, Reference Position. Article name, authors, DOI and location where the information was found.

Table 12: Dataset statistics.

| | |
|---|---|
| Experiments | 68 110 |
| Unique pairs | 46 717 |
| Unique sequences | 1237 |
| Unique molecules | 596 |
| Species | 11 |

**Data preparation.**   Some OR-molecule pairs may be tested in several different experiments with possibly different outcomes. In order to train and evaluate our model, we extracted unique pairs from the dataset as follows: If a given pair has a dose-response experiment, we take its result and ignore others. Otherwise, if a pair has multiple measurements but not a full dose-response curve, we consider it as a secondary screening and we take this pair and its response only if there is no conflict between the experiments. If there is a conflict, we discard the pair entirely (in total 166 pairs were discarded).

