# OpenReview forum: "Matching receptor to odorant with protein language and graph neural networks"
_ICLR.cc/2023/Conference — ICLR 2023 poster_

### Official Review · Reviewer_21XM · 2022-10-21

**Confidence:** 3
**Correctness:** 3
**Technical Novelty And Significance:** 3
**Empirical Novelty And Significance:** 2
**Recommendation:** 6

**Clarity, Quality, Novelty And Reproducibility:**

Clarity: Clear presentation of the problem at hand, the method, and the results.

Originality: New method for predicting protein-molecule interaction.

Quality: The empirical evaluation is focused on odor perception and might be of limited interest to the wider community.

**Strength And Weaknesses:**

Strengths:
+ Protein-molecule interaction is generally a problem of interest to many communities.
+ The proposed method is an interesting way to combine a protein language model (for the protein) and a GNN (for the molecule), and this method might benefit other problems outside of odor perception.

Weaknesses:
+ The empirical evaluation is focused on odor perception.
+ While the proposed method is more general according to the authors, the empirical performance does much some of the existing methods (Table 4).

**Summary Of The Paper:**

Predicting protein-molecule interaction by combining a pre-trained protein language model and a GNN. The method was evaluated empirically in the context of odor perception.

**Summary Of The Review:**

Interesting method for predicting protein-molecule interaction, with empirical evaluation very focused on a subdomain.

---

> ### Author Response · Authors · 2022-11-18
> **Response to reviewer 21XM**
>
> We thank the reviewer 21XM for his favorable review and for his careful reading of the manuscript. We hope that the additional experiments and the following responses will show that we have made serious efforts to answer and address each of the issues.
>
> **Comment 1**: *The empirical evaluation is focused on odor perception.*
>
> We agree with the reviewer that broader application is an important issue that needed to be addressed. We made additional experiments where we test our model on two other datasets, namely KIBA [8] and DAVIS [9] and we compare the results to other SOTA models for drug target interaction (DTI) prediction HyperAttentionDTI [6] and MolTrans [7] which were reported to be the best on the two datasets by [6]. Overall, HyperAttentionDTI leads to best AveP for KIBA and DAVIS and our approach is comparable to MolTrans on KIBA. On the OR ligand prediction (our data), which is the main focus of this work, and yet unresolved and somewhat overlooked task, our model outperforms both DTI models.
>
> It is worth noting that the statistical properties of KIBA, DAVIS and OR-ligand are different (see Table 10 in revised manuscript). In OR-ligand there are 1237 unique sequences, 596 unique molecules, and distributions of number of tests per receptor and per molecule are highly imbalanced (see Fig. 7 in Supplementary material). Out of 1237 sequences 527 sequences have no positive sample (i.e. no known active compound). On the other hand, in section E.2. in revised manuscript we report the distributions for KIBA and DAVIS, which are both kinase inhibition datasets. There are 225 sequences and 2068 molecules in KIBA and only 5 sequences do not have any positive example. In DAVIS with 379 sequences and 68 molecules, all possible 25772 pairs were tested. The structure of tested molecules in OR-ligand also differs from those in KIBA and DAVIS. We include a comparison with the MolTrans and HyperAttentionDTI models in Table 1 and tests on the KIBA and DAVIS datasets in Supplementary material E.2.
>
> **Comment 2**: *While the proposed method is more general according to the authors, the empirical performance does much some of the existing methods (Table 4).*
>
> We are thankful for the reviewer’s critical point. Our model outperforms all other approaches for OR ligand prediction task except for Kowalewski et. al. [13] in AUROC metric. However, Kowalewski et.al. fit one separate model for each OR and their approach is the only one that can not be applied to any other olfactory receptor than those which have enough known active compounds. Thus, the task of Kowalewski et. al. avoids the major challenges of scarce data and orphan receptors and is easier than the general problem of predicting response for any OR-molecule pair. To support this claim we made a small experiment by excluding sequences with 2 or less known active compounds from the test set. When we test our original model on these new test sets (827 pairs on average) we gain 8\% performance (MCC $0.652$ compared to $0.605$ in the case of the full test set) without any change to the model training. Thank you for pointing this out and we clarified the difference between Kowalewski et. al. and other approaches in the manuscript in Section 5.3.
>
> **References:**
>
> [6] Qichang Zhao, Haochen Zhao, Kai Zheng, and Jianxin Wang. HyperAttentionDTI: improving drug-protein interaction prediction by sequence-based deep learning with attention mechanism. Bioinformatics, 38(3):655–662, October 2021.
>
> [7] Kexin Huang, Cao Xiao, Lucas M Glass, and Jimeng Sun. MolTrans: Molecular Interaction Transformer for drug–target interaction prediction. Bioinformatics, 37(6):830–836, 10 2020
>
> [8] Jing Tang, Agnieszka Szwajda, Sushil Shakyawar, Tao Xu, Petteri Hintsanen, Krister Wennerberg, and Tero Aittokallio. Making sense of large-scale kinase inhibitor bioactivity datasets: a comparative and integrative analysis. J. Chem. Inf. Model., 54(3):735–743, March 2014.
>
> [9] Mindy I Davis, Jeremy P Hunt, Sanna Herrgard, Pietro Ciceri, Lisa M Wodicka, Gabriel Pallares, Michael Hocker, Daniel K Treiber, and Patrick P Zarrinkar. Comprehensive analysis of kinase inhibitor selectivity. Nat. Biotechnol., 29(11):1046–1051, October 2011.
>
> [13] Joel Kowalewski and Anandasankar Ray. Predicting human olfactory perception from activities of odorant receptors. iScience, 23(8):101361, 2020. ISSN 2589-0042.

---

### Official Review · Reviewer_zko6 · 2022-10-23

**Confidence:** 4
**Correctness:** 4
**Technical Novelty And Significance:** 3
**Empirical Novelty And Significance:** 3
**Recommendation:** 8

**Clarity, Quality, Novelty And Reproducibility:**

The overall quality of the work is generally high: the data set collected is large and the model architecture is sophisticated, showing considerable thought.

The clarity of the question being asked is good but some of the methodological details are lacking in clarity; see description above.

The paper approaches a question that has been asked by a range of recent papers, but uses a novel architecture (albeit one that has similarities to architectures used for other similar problems in chemistry); I personally felt that especially the key usage of attention to model non-bonded interactions was quite original. The approach of relating their predictions to olfactory perception - while building on existing models of olfactory perception - was also quite original to the best of my knowledge.


**Strength And Weaknesses:**

This work represents an important step toward models that can reliably predict protein-ligand interactions, focused on the very interesting case of odorant receptors and small molecules. The architecture presented is one that is original and sophisticated, and in Section 5.1 the authors make a convincing case that its choices are not arbitrary. While it is not the first paper to use AI to predict olfactory recognition, previous studies have used much simpler models that rely on SMILES encodings of small molecules; I think especially that the attention mechanism learned over the molecular graphs to learn nonbonded interactions strikes me as an important methodological innovation in this study.

The “Generalization” section of the manuscript is very important, and the authors perform a number of important statistical tests: notably to not only perform random train/test splits of receptors and molecules but also to cluster them, such that members of the test set are relatively different from the training set. Their models perform quite poorly in these situations, and the authors are honest about this case. I personally find it to be a strength of the paper that they did this test, although it obviously opens the door to future improvements of the model.

The dataset generated of over 45,000 OR-olfactant pairs is also a strength of this paper and will serve as an invaluable reference for future researchers in the field.

Broadly, I feel that the model is stronger at modeling small molecule space than protein space, especially as the protein sequence representation relies entirely on the pretrained [CLS] tokens of ProtBERT. I would personally recommend trying a few variations on this, although I wouldn’t say that they are required because I understand that they can be computationally intensive and challenging to add into such an already large model; I would recommend this as a direction of future research (which the authors also note!)

I also think that some technical parts of the paper are somewhat unclear and could use slightly more description. For example, several aspects of their architecture are not well explained; the message passing neural networks (Q-MPNN and KV-MPNN) are never mentioned in the main text - are these the “two separate identical GNNs for queries and keys/values” mentioned? If, so that should be made clear, along with what the “Q” and “KV” mean. The meaning of “X_old”, “X_new” and “E” in Figure 1 are also not described. I can guess as to their meaning, but it would be better for it to be clearly stated in the legend. Explaining this terminology will also make the “ablations” section more clear, although there are also a few architectures introduced first in that section (e.g. GAT) that could be introduced slightly more clearly and justified in their use.

In Section 6, the precise definition of “combinatorial code” for each odorant is also somewhat unclear - I am guessing that they mean that they are the predicted recognition of the odorant by each of the 385 human ORs in the model presented earlier, but this is not explicitly stated. Therefore, while they are very mathematically precise about how their clustering score is defined, I initially didn’t understand what actual data they were analyzing.


**Summary Of The Paper:**

The authors present a novel approach for identifying which small molecules will be recognized by different mammalian olfactory receptors. While similar approaches have been presented previously, this study introduces a novel combination of neural network architectures to model the protein-ligand interaction, relying on the learned representation of protein sequences in the ProtBERT language model paired with a graph neural network representation of the small molecule with an attention mechanism to learn non-bonded interactions. They show that this architecture is able to outperform pre-existing techniques at predicting interactions within a newly collected database of 46,450 odorant-OR pairs and dissect how different variations on the network leaving out aspects of the model lead to worse performance at these benchmarks.

**Summary Of The Review:**

This paper is overall an impressive piece of work and I think that it would be a valuable addition to ICLR. The architecture used in particular is quite novel and an important increase in sophistication and predictive power over other architectures used in the field, although some elements of its design and testing could be made somewhat more clear in the paper. Especially if these aspects are stated more explicitly in the paper, I would strongly recommend this paper for acceptance.

---

> ### Author Response · Authors · 2022-11-18
> **Response to reviewer zko6**
>
> We thank reviewer zko6 for his very favorable review of our work. We believe that this work and the new dataset will help push the boundaries of the olfaction research. Additionally, thanks to various information that are gathered in the dataset about data quality, type of bio-assays or experimental conditions such as screening concentration it may be used to guide experimentalists in their decisions for the most appropriate test conditions or used by researches in other fields as an interesting real-life benchmark for phenomena like label noise.
>
> **Comment 1**: *Broadly, I feel that the model is stronger at modeling small molecule space than protein space, especially as the protein sequence representation relies entirely on the pretrained [CLS] tokens of ProtBERT. I would personally recommend trying a few variations on this, although I wouldn’t say that they are required because I understand that they can be computationally intensive and challenging to add into such an already large model; I would recommend this as a direction of future research (which the authors also note!).*
>
> We fully agree with the reviewer’s comment. Indeed, in near future we plan to explore other protein embeddings. In particular we believe that using MSA-based embedding such as one from MSA Transformer [5] or Evoformer block from AlphaFold 2 [12] are a promising direction. These models do not output an aggregated representation of the protein and we plan to explore computationally more demanding models that explicitly represent amino acids. We plan to build on the results presented in this manuscript regarding molecule processing, mixing strategy as well as data biases and their treatment.
>
> **Comment 2**: *I also think that some technical parts of the paper are somewhat unclear and could use slightly more description. For example, several aspects of their architecture are not well explained; the message passing neural networks (Q-MPNN and KV-MPNN) are never mentioned in the main text - are these the “two separate identical GNNs for queries and keys/values” mentioned? If, so that should be made clear, along with what the “Q” and “KV” mean. The meaning of “X\_old”, “X\_new” and “E” in Figure 1 are also not described. I can guess as to their meaning, but it would be better for it to be clearly stated in the legend. Explaining this terminology will also make the “ablations” section more clear, although there are also a few architectures introduced first in that section (e.g. GAT) that could be introduced slightly more clearly and justified in their use.*
>
> *In Section 6, the precise definition of “combinatorial code” for each odorant is also somewhat unclear - I am guessing that they mean that they are the predicted recognition of the odorant by each of the 385 human ORs in the model presented earlier, but this is not explicitly stated. Therefore, while they are very mathematically precise about how their clustering score is defined, I initially didn’t understand what actual data they were analyzing.*
>
> Thank you for taking an effort to make the manuscript as clear as possible for the reader and we did our best to clarify all the points. We stated the definition of Q-MPNN and KV-MPNN on page 3 and the meaning of X\_old, X\_new and E in Fig. 1. We also added justification for using GAT as a baseline on page 5. At the beginning of Section 6 we added definition of the combinatorial code and revised sentence explaining the aim of the section. We hope that these changes will make the goal of the section more clear to the reader.
>
>
> **References:**
>
> [12]  John Jumper, Richard Evans, Alexander Pritzel, Tim Green, Michael Figurnov, Olaf Ronneberger, Kathryn Tunyasuvunakool, Russ Bates, Augustin Žídek, Anna Potapenko, Alex Bridgland, Clemens Meyer, Simon A. A. Kohl, Andrew J. Ballard, Andrew Cowie, Bernardino Romera-Paredes, Stanislav Nikolov, Rishub Jain, Jonas Adler, Trevor Back, Stig Petersen, David Reiman, Ellen Clancy, Michal Zielinski, Martin Steinegger, Michalina Pacholska, Tamas Berghammer, Sebastian Bodenstein, David Silver, Oriol Vinyals, Andrew W. Senior, Koray Kavukcuoglu, Pushmeet Kohli, and Demis Hassabis. Highly accurate protein structure prediction with alphafold. Nature, 596(7873):583–589, Aug 2021.

---

### Official Review · Reviewer_pDHK · 2022-10-23

**Confidence:** 3
**Correctness:** 3
**Technical Novelty And Significance:** 2
**Empirical Novelty And Significance:** 3
**Recommendation:** 5

**Clarity, Quality, Novelty And Reproducibility:**

The text is clear and the code for the model source is available.


**Strength And Weaknesses:**

Positives:
- Tackling a new and interesting problem of predicting receptor-odorant binding.
- Proposing a new model involving GNN and BERT embedding for olfactory receptors.
- Thorough ablation study of the model.
- Gathering a new dataset for olfactory receptors.

Negative:
- The so-called “localization” strategy of adding protein embeddings to the graph, while seemingly improving prediction, does not give any physical interpretation of the underlying binding mechanism of receptors to the odorants.
- Why should all the codes of the graph have the share of the protein embedding? Intuition says the docking site should play a more significant role than the rest of the molecule.
- The choice of [CLS] as the embedding is not well-justified. Have the authors done ablation studies over other types of protein embedding for example ESM?
- Most of the empirical studies are ablation type and comparisons to standard methods are very limited.
- How does the model compare with the recently developed SE(3) invariant deep learning models of molecules?
- It is intuitive that the docking site of the protein is more important than the rest. Instead of a global protein embedding can the model incorporate this locality?


**Summary Of The Paper:**

The paper develops a new model for olfactory receptors.

**Summary Of The Review:**

Overall the paper looks at an interesting problem however the model falls short in some key aspects as described which can be improved upon.

---

> ### Author Response · Authors · 2022-11-18
> **Response to reviewer pDHK**
>
> We would like to thank the reviewer pDHK for providing helpful and insightful comments and for the careful attention that he has paid to our manuscript. A number of issues and questions were raised that we have addressed in the revised version. Below are specific changes that we have made in response to each comment.
>
> **Comment 1**: *The so-called “localization” strategy of adding protein embeddings to the graph, while seemingly improving prediction, does not give any physical interpretation of the underlying binding mechanism of receptors to the odorants.*
>
> Thank you for raising this interesting point. Indeed, the main goal of the "localization" strategy is to enhance the predictive power of the model by observing the need for locally-changed protein representation. Binding site of olfactory receptors has been thoroughly investigated (more than 30\% of the OR sequences have been subjected to point mutations in a number of publications, see references within [1]). In summary, they represent more than 100 distinct positions within the sequence of OR. Thus, nowadays the binding site is well located. However, despite the knowledge of the binding site, finding new active compounds for ORs and eventually revealing the combinatorial coding of odorant molecules remains a major challenge.
>
> If one aims at atomistic description of the activation process then other well-developed computational approaches such as docking or molecular modelling can be applied on a predicted active pair. In principle one can also extract specific patterns from the analysis of attention heads and give a rationale of a specific receptor-molecule interaction. Although protein representation is aggregated to the [CLS] token, the molecular topology is accessible in the model and one can identify receptor-specific binding groups in the molecule.
>
> **Comment 2** : *Why should all the codes of the graph have the share of the protein embedding? Intuition says the docking site should play a more significant role than the rest of the molecule.*
>
> We agree that the docking site of a receptor is important for the activation process as this specific region of the protein is dedicated to the ligand binding. However, it has been demonstrated for olfactory receptors that the response to an odorant can be drastically altered by mutations even if they are distant from the binding pocket [2, 1]. As for GPCR [3], a highly subtle allosteric network is responsible for OR activation mechanism. Thus, even though binding event is an important part of the activation process, it is not sufficient alone for observing response in *in-vitro* experiments. Since we aim at predicting the activation of an OR (i.e. the result of an *in-vitro* test) we allow the model to incorporate other protein information necessary for the activation, such as molecular switches [3], by taking into account the full protein sequence.
>
> As we have no prior information which binding group (or in general substructure in the molecule) plays a significant role in the activation process, we allow access to the full protein embedding for all nodes in the graph.
>
> **References:**
>
> [1] Claire A de March, Jeremie Topin, Elise Bruguera, Gleb Novikov, Kentaro Ikegami, Hiroaki Matsunami, and Jerome Golebiowski. Odorant receptor 7D4 activation dynamics. Angew. Chem. Int. Ed Engl., 57(17):4554–4558, April 2018.
>
> [2] Claire A. de March, Yiqun Yu, Mengjue J. Ni, Kaylin A. Adipietro, Hiroaki Matsunami, Minghong Ma, and Jerome Golebiowski. Conserved residues control activation of mammalian g protein-coupled odorant receptors. Journal of the American Chemical Society, 137(26):8611–8616, 2015
>
> [3] Alexander S Hauser, Albert J Kooistra, Christian Munk, Franziska M Heydenreich, Dmitry B Veprintsev, Michel Bouvier, M Madan Babu, and David E Gloriam. GPCR activation mechanisms across classes and macro/microscales. Nat. Struct. Mol. Biol., 28(11):879–888, November 2021.

---

> ### Author Response · Authors · 2022-11-18
> **Response to reviewer pDHK part 2**
>
> **Comment 3**: *The choice of [CLS] as the embedding is not well-justified. Have the authors done ablation studies over other types of protein embedding for example ESM?*
>
> We appreciate the reviewer’s comment and provide rationale to our choice of protein embedding model. The embedding choice was based on the work done by [4] where the authors conducted experiments using protTrans models. They showed that these models can learn structure in their attention heads and some of the heads specifically focus on the binding sites. ESM-1b is a very similar model to protBERT where the difference between the two is that ESM-1b is a Transformer with 33 layers compared to 30 layers of protBERT. Both models are trained in a BERT masked-language modeling setting.
>
> In light of time constraints we did not run ablation on ESM embeddings, but we expect that the performance would be very similar to using ESM-1b embedding. However, we fully agree with the reviewer that the protein representation could be enhanced, but as the reviewer zko6 pointed out, incorporating different embedding is challenging and may be computationally intensive. We plan to investigate different protein embeddings in the future where we would like to explore MSA-based embedding (e.g. MSA Transformer or Evoformer block of AlphaFold) which were shown to outperform single-sequence models [5]. Working with multiple sequences, these models do not provide aggregated representation of the sequence and one needs to define a suitable aggregation method of the amino acid representations or a new way of combining the molecular and protein embeddings.
>
> **Comment 4**: *Most of the empirical studies are ablation type and comparisons to standard methods are very limited.*
>
> We added additional experiments comparing to other 2 SOTA drug target interaction prediction models [6] and [7], which based on [6] perform best on the two kinese datasets (KIBA [8] \& DAVIS [9]). Overall our model outperforms all the other approaches on the OR activation prediction. In the revised manuscript we added the comparisons to Table 1.
>
> **Comment 5**: *How does the model compare with the recently developed SE(3) invariant deep learning models of molecules?*
>
> Thank you for your insightful suggestion. As we noted in the manuscript (in Section 3) we do not use 3D coordinates of a molecule nor a receptor and the molecular representation is purely abstract. Therefore, translations and rotations are not defined in this abstract space and we can not straightforwardly use 3D-based models. However, to further expand the set of models that we compare our results to, we added performance of Graph isomorphism network (GIN) [10] on top of the 2 SOTA models for drug-target interaction predictions  (results are presented in Table 1).
>
> **References:**
>
> [4] Jesse Vig, Ali Madani, Lav R. Varshney, Caiming Xiong, richard socher, and Nazneen Rajani. BERTology meets biology: Interpreting attention in protein language models. In International Conference on Learning Representations, 2021
>
> [5] Roshan Rao, Jason Liu, Robert Verkuil, Joshua Meier, John Canny, Pieter Abbeel, Tom Sercu, and Alexander Rives. Msa transformer. In Marina Meila and Tong Zhang, editors, Proceedings of the 38th International Conference on Machine Learning, volume 139 of Proceedings of Machine Learning Research, pages 8844–8856. PMLR, 18–24 Jul 2021
>
> [6] Qichang Zhao, Haochen Zhao, Kai Zheng, and Jianxin Wang. HyperAttentionDTI: improving drug-protein interaction prediction by sequence-based deep learning with attention mechanism. Bioinformatics, 38(3):655–662, October 2021.
>
> [7] Kexin Huang, Cao Xiao, Lucas M Glass, and Jimeng Sun. MolTrans: Molecular Interaction Transformer for drug–target interaction prediction. Bioinformatics, 37(6):830–836, 10 2020
>
> [8] Jing Tang, Agnieszka Szwajda, Sushil Shakyawar, Tao Xu, Petteri Hintsanen, Krister Wennerberg, and Tero Aittokallio. Making sense of large-scale kinase inhibitor bioactivity datasets: a comparative and integrative analysis. J. Chem. Inf. Model., 54(3):735–743, March 2014.
>
> [9] Mindy I Davis, Jeremy P Hunt, Sanna Herrgard, Pietro Ciceri, Lisa M Wodicka, Gabriel Pallares, Michael Hocker, Daniel K Treiber, and Patrick P Zarrinkar. Comprehensive analysis of kinase inhibitor selectivity. Nat. Biotechnol., 29(11):1046–1051, October 2011.
>
> [10] Keyulu Xu, Weihua Hu, Jure Leskovec, and Stefanie Jegelka. How powerful are graph neural networks? In International Conference on Learning Representations, 2019.

---

> ### Author Response · Authors · 2022-11-18
> **Response to reviewer pDHK part 3**
>
> **Comment 6**: *It is intuitive that the docking site of the protein is more important than the rest. Instead of a global protein embedding can the model incorporate this locality?*
>
> We are grateful for the reviewers’ careful reading. As we pointed out in the response to the second question, the amino acids outside of the docking site of an OR also play an important role in the activation process. Therefore, it's relevant from a biological point of view to take into account the full protein sequence in the model.
>
> Still, if one represents just the docking site as a fixed sized vector, the model can take it as a protein input. However, one must be cautious because simply taking a subpart of the sequence corresponding to the binding site and using it as an input to BERT would probably result in an incorrect representation. The fold of the sub-sequence would be entirely different from the fold of the original protein binding site and under the assumption that BERT is an unsupervised structure learner (as shown by [11, 4]) the [CLS] token may contain misleading information.
>
> **References**:
>
> [11] Roshan Rao, Joshua Meier, Tom Sercu, Sergey Ovchinnikov, and Alexander Rives. Transformer protein language models are unsupervised structure learners. In International Conference on Learning Representations, 2021.

---

### Author Response · Authors · 2022-11-18
**Revision Summary**

We are grateful to our reviewers and we would like to thank them for their work in reviewing the manuscript. Their precise and sharp comments on the model we built to decipher the combinatorial code of olfactory receptor activation allowed us to complete our results, to reinforce our conclusions, as well as improve the clarity of the manuscript. In this revised work, we provide answers to both technical and fundamental questions about model performance, protein and molecule modeling choices and answer point by point to the reviewers' remarks. In particular, in the revision, we have evaluated the performance of our method on other datasets (KIBA \& DAVIS). We also compared the performance of recently developed methods (MolTrans & HyperAttentionDTI) on our OR-molecule data. These points were raised by the reviewers pDHK and 21XM. Modifications are highlighted in blue in the revised manuscript.

---

### Decision · Program_Chairs · 2023-01-20

**Decision:**

Accept: poster

**Justification For Why Not Higher Score:**

not sufficient novelty

**Justification For Why Not Lower Score:**

interesting results

**Metareview: Summary, Strengths And Weaknesses:**

The paper addresses a classically difficult problem of olfactory prediction, using olfactory receptor-small molecule binding prediction. Compared to previous approaches that considered mainly the small molecule in a standalone fashion, the proposed approach also incorporates information about the structure of the protein receptor. Another contribution is a new dataset. We believe the paper merits publication.

**Note From Pc:**

if the above contains the word "oral" or "spotlight" please see: "oral" presentation means -> notable-top-5% and "spotlight" means -> notable-top-25%. As stated in our emails, we are disassociating presentation type from AC recommendations